# Fabrication of Temperature Sensors with High-Performance Uniformity through Thermal Annealing

**DOI:** 10.3390/ma16041491

**Published:** 2023-02-10

**Authors:** Hongrui Liu, Yongchun Li, Weiji Xie, Xinyi Zhou, Jishuang Hong, Junfeng Liang, Yanghui Liu, Wei Li, Hong Wang

**Affiliations:** 1School of Materials, Shenzhen Campus of Sun Yat-sen University, Shenzhen 518107, China; 2Department of Anesthesiology, Sun Yat-sen University Cancer Center, State Key Laboratory of Oncology in South China, Guangzhou 510060, China

**Keywords:** conductive composite, temperature sensor, annealing, high uniformity, body temperature

## Abstract

It is considered to be of great significance to monitor human health and track the effect of drugs by measuring human temperature mapping through flexible temperature sensors. In this work, we found that the thermal annealing of flexible temperature sensors based on graphite–acrylate copolymer composites can not only improve the temperature coefficient of resistance (TCR) values of the devices, but also greatly improve the uniformity of the performance of the devices prepared in parallel. The best results were obtained when the devices were annealed at 100 °C, which is believed to be due to the rearrangement of graphite particles to generate more uniform and numerous conductive channels within the conductive composite. We believe this finding might promote the practical development of flexible temperature sensors in body temperature sensing for health maintenance and medical applications.

## 1. Introduction

Body temperature is highly related to numerous human diseases; the occurrence and cure of diseases can be revealed by the temperature change curve [1,2]. Thus, immediately and accurately monitoring the temperature in specific parts of the human body is of great significance for health management and the study of the effect of drugs [3]. Human skin transmits temperature due to internal blood flow, and the temperature of the torso is mainly maintained at about 33–42 °C with a minor change [4,5,6]. Therefore, temperature sensors applied to monitor the human body are required to have high-accuracy results regarding body temperature. Furthermore, the measurement at a single point can only obtain the temperature at the measuring position, which is incapable of accurately locating the lesion site and drug action point [7,8]. Additionally, the diversity of individual body temperatures further increases the complexity of medical judgment and treatment through single-point measurements [9,10]. The development of a temperature sensor array that can accurately measure the temperature of the human body is still a great challenge.

A conventional mercury thermometer has a long response time and can only measure the temperature at a single point, which cannot meet the requirements at present of precision medicine [11,12]. An infrared thermometer can measure the temperature of the human body’s surface in real time, but it is expensive [13,14,15] and can only measure the temperature of the exposed body surface. Furthermore, the measurement results are easily affected by the environment [15,16,17]. In recent years, flexible temperature sensors have attracted extensive attention due to their advantages, e.g., low cost, flexibility, and implantable nature. Ultra-thin metal structures, such as Pt metal films [18], Ag nanocrystal films [19], Cu or Ag nanowires [20], and Au serpentine wires [21], with good stability and wide linear range were constructed on flexible polymer substrates to prepare a variety of flexible temperature sensors, which can be applied in intelligent prosthetics, skin-coupled hyperthermia patches, daily temperature measurements, and skin-surface temperature surveys caused by air and dripping. However, the temperature coefficient of resistance (TCR) value, which describes the relationship between resistance R and temperature T, i.e., $\alpha =\frac{\left(\Delta R/R\right)}{\Delta T}\;$, is generally small, resulting in the requirement of complex external amplification circuits for practical measurement [22]. Additionally, those temperature sensitive materials might generate microcracks after multiple bends, which might increase DC resistances and shorten the service life of corresponding devices [23]. Wang et al. developed a fully printed, flexible temperature sensor based on cross-linked poly(3,4-ethylenedioxythiophene):poly(styrene sulfonate) (PEDOT:PSS), and integrated it into a printed flexible hybrid circuit to obtain a wireless temperature-sensing platform that can perform stable real-time health monitoring, which also has the disadvantage of a low TCR value [24]. In this case, conductive composites have attracted significant attention owing to their ultra-high TCR value. Most of the conductive composites have a positive temperature coefficient (PTC) effect, and when the temperature rises, the polymer matrix has a higher thermal expansion rate than that of the conductive fillers, resulting in a decrease in internal conductive channels and an increase in resistance [25]. Especially, conductive composite temperature sensors based on semi-crystalline polymers exhibit extreme sensitivity to temperature, where the resistance changes by several orders of magnitude when the temperature fluctuates within 4 °C [26,27]. The sudden change in resistance is generally believed to be due to the fact that the dispersion of conductive fillers in the polymer matrix increases is rapidly caused by the transformation of crystalline zones into amorphous ones during the melting of semi-crystalline polymers, where the conductive fillers are considered to be only exist in the amorphous regions [28].

Yokota et al. developed a new kind of semi-crystalline polymer called an acrylate copolymer (AC), whose melting point can be adjusted by regulating the ratio of the two monomers. By using the conductive composite fabricated by mixing graphite (Gr) fillers with AC, they successfully prepared a temperature sensor with a high accuracy of 0.1 °C near the human-body-temperature level. Temperature sensor arrays based on the graphite–acrylate copolymer (Gr–AC) was also developed, which can be applied to the measurement of temperature maps of a rat lung [29]. Yang et al. also used the material to prepare miniaturized respiration-detection arrays, which effectively detected the respiration rates of humans at a distance of 30 cm [30].

However, most of the previous works mainly focus on improving the performance of a single device, while how to improve the uniformity between devices in an array is rarely addressed. In this work, we can observe that the uniformity as well as TCR values of the temperature sensors based on a Gr–AC conductive composite can be improved by thermal annealing. When annealed at 100 °C for 30 s, the temperature sensors have the best performance. Annealing can induce the aggregation of dispersed conductive fillers to form numerous and stable conductive-channel flexible temperature sensor arrays with high-precision behavior for the determination of human-body-temperature maps.

## 2. Experimental Section

*Synthesis of acrylate copolymer:* Butyl acrylate (BA, 99%), anhydrous tetrahydrofuran (THF, 99%), and 2,2-dimethoxy-2-phenylacetophenone (DMPA, 99%) were purchased from Aladdin. Octadecyl acrylate (OA, 97%) was provided by Sigma. Graphite powder with a diameter of 2–3 μm was provided by 3A Materials. All the reagents were used without further purification. BA and OA were mixed in anhydrous THF solvent, DMPA was added as a photo initiator, and polymerization occurred in an anhydrous and oxygen-free environment under ultraviolet lamp irradiation for 24 h. The THF solvent was removed by pumping to obtain the acrylate copolymer.

*Preparation of Gr–AC conductive composite:* The acrylate copolymer and graphite powders were magnetically stirred at 70 °C for 24 h, according to the mass ratio of 5:2. Subsequently, the mixture was stirred further by a planetary centrifugal vacuum mixer (Thinky MZ-8) for 5 min.

*Device Fabrication*: The interdigital electrodes were prepared on a 25 μm thick polyimide (PI) film by depositing 5 nm Cr as an adhesion layer, followed by 30 nm of Au through a mask. Then, the Gr–AC conductive composite was hot-pressed onto the Au electrode. The temperature-testing band containing 2 × 9 devices was fabricated by screen-printing Gr–AC on the circuits, where the circuits were purchased from Shenzhen Shenya Precision Circuits Co., Ltd., Shenzhen, China.

*Materials and Device Characterization*: Differential scanning calorimetry (DSC) measurements were performed using PerkinElmer DSC 6000. Scanning electron microscope (SEM) images were obtained by ZEISS Sigma 300. X-ray diffraction (XRD) measurements were performed using the TD-3500 (Dandong Tongda Science and Technology, Dandong, China) X-Ray diffractometer system. The basic electrical properties were tested by Keithley 2450.

## 3. Results and Discussions

Figure 1a shows the schematic diagram (upper part) and photograph (lower part) of a single temperature sensor. Firstly, the semi-crystalline acrylate copolymer (AC) was synthesized by using octadecyl acrylate (OA) and butyl acrylate (BA) as precursors, 2,2-dimethoxy-2-phenylacetophenone (DMPA) was used as the photo initiator, and the reaction was conducted in an oxygen-free environment under the irradiation of 365 nm of ultraviolet light. Then, graphite powders with a diameter of 2–3 μm were mixed with an AC polymer in the proportion of 40 wt% to prepare the Gr–AC conductive composite. When the proportion of OA is 58 mol%, the melting point of the composite is 38.5 °C after the addition of Gr, as shown in Figure 1b. In addition, the AC and Gr–AC composite were characterized by X-ray diffraction (XRD), and the results are shown in 1c. A peak located at 22° was observed in the scan of the AC sample, consistent with the results reported in the literature [29], which is considered as the diffraction of crystalline AC. After incorporating graphite, a new peak at 26° identified as crystalline Gr(002) appeared [31]. Then, the obtained conductive composite was mounted on the pre-designed interdigital electrodes on a 25 μm polyimide (PI) film, where the electrode width and channel length were 100 and 130 μm, respectively.

The resistance of the composite can be obtained by measuring the I–V curve, and the value was calculated to be 2.7 × 10^4^ Ω at room temperature (see Appendix A). The resistance gradually increases with the increase in temperature. Since the temperature of the mean skin temperature of humans is generally not lower than 30 °C [32], we mainly studied the relationship between resistance and temperature in the range from 30 °C to a temperature before the material melted. In this work, a maximum temperature of 35.5 °C was chosen to ensure that all resistances were within the measuring range of the instrument, since the upper limit of the resistance measurement of the source meter we used was 2 × 10^8^ Ω. In order to study the uniformity of the devices based on the Gr–AC composite, 18 devices were prepared in parallel, and then their resistance versus temperature curves were tested, respectively. The average value of the measured results is plotted in Figure 1d. The TCR value from 30 °C to 35.5 °C was calculated to be 12.6 °C^−1^, which is equivalent to the sensitivity of the device prepared by using Gr–AC in the literature [30,33], and the corresponding device sensitivity is much higher than that of the sensors based on polydimethylsiloxane (PDMS) [34], polyurethane (PU) [35], and poly(3,4-ethylenedioxythiophene):poly(styrene sulfonate) (PEDOT:PSS) [36]. In order to improve the uniformity of these devices, we annealed them 100 °C and tested the resistance–temperature curves again, after they were restored to room temperature. In order to ensure the complete melting of the Gr–AC composites, the annealing time was set to 30 s, while a longer annealing time might not improve the performance of the corresponding temperature sensors further (See Appendix A). The results show that the average TCR value increase to 43.2 °C^−1^, and the uniformity of the devices is significantly improved.

To study the influence of annealing temperature on the devices to further improve the device’s performance and uniformity, we analyzed the fluctuation of the devices when they were annealed at various temperatures. Figure 2a plots the temperature dependence of resistance of the devices after annealing at 50, 75, 100, 125, and 150 °C for 30 s, where the TCR values in the range of 30 to 35.5 °C were calculated to be 21.6, 39.7, 43.2, 33.3, and 28.1 °C^−1^, respectively. The TCR achieved the highest value when the devices were annealed at 100 °C, which can endow the devices the highest sensitivity. However, increasing the annealing temperature further cannot increase the TCR value, but causes the TCR value to gradually decrease. On the other hand, the uniformity presents a similar trend with the increase in the annealing temperature. Figure 2b shows that the variation in the resistance fluctuation decreases with the increase in the annealing temperature until it reaches 100 °C, while increasing the annealing temperature further might cause the resistance fluctuation to start increasing again. Therefore, we could obtain the highest TCR value as well as the best uniformity when the devices were annealed at 100 °C for 30 s.

To understand the effect of the annealing temperature on the devices, a cross-section of the Gr–AC composite was characterized by scanning electron microscope (SEM) to analyze the distribution of Gr particles in AC, and the results are shown in Figure 3. Figure 3a shows the SEM image of the as-prepared sample, where the white regions are AC polymers and the black ones with layered structures are Gr particles. Benefitting from the long stirring time during the preparation process, the graphite particles were uniformly distributed in the AC. Figure 3b–f show the SEM images of the samples annealed at 50–150 °C, respectively. With the increase in the annealing temperature, the Gr particles began to partially aggregate in the AC, where the higher the temperature is, the more serious the aggregation of the Gr particles. Meanwhile, the formation of a high number of aggregates was accompanied by the interconnection of the conductive fillers to form short chains, which can serve as new conductive channels in the composite. In order to analyze and compare the distribution of Gr particles in AC more accurately, we calculated the proportion of Gr in each image using MATLAB software, according to the different gray values for Gr and AC in the SEM images. The specific analysis method and results are shown in Appendix A. The percentages of Gr in the samples before and after annealing at 50–150 °C were calculated to be 62.5%, 63.9%, 63.4%, 65.4%, 64.1%, and 60.2%, respectively. Since the actual content of Gr in every sample was consistent, the change in the Gr proportion was due to the difference in the aggregation and interconnection degrees of Gr. The sample annealed at 100 °C presented the largest Gr proportion, while the one annealed at 150 °C had the smallest Gr proportion. That is because the graphite particles show a greater contrast in the image after aggregation. However, if the annealing temperature is too high, a high number of aggregated particles connect with each other, which exposes more of the AC matrix in the cross-section in the SEM image, thus reducing the measured Gr proportion value.

Since AC is a kind of semi-crystalline polymer, the changes in the Gr–AC performance caused by annealing can be explained by the thermal expansion model [37]. Figure 4 shows schematic images to estimate the evolution of Gr distribution in AC during annealing. Figure 4a shows the internal state of the unannealed sample, where the gray AC matrix is partially crystallized, black Gr particles are dispersed in the amorphous region, and the highlighted parts represent possible conductive channels formed by Gr particles. When the temperature rises, as shown in Figure 4b, the crystalline area of AC disappears and the composite becomes a liquid. The graphite particles rearranged due to thermal movement and began to aggregate. After the heating was stopped, as shown in Figure 4c, the composite rapidly cooled down, and the area without graphite also rapidly crystallized, while the area containing Gr retained an amorphous state, so as to fix the graphite movement and aggregation caused by thermal annealing. The shrinkage of the polymer’s volume after cooling caused these aggregated Gr particles to form new, conductive channels. A high annealing temperature might reduce the viscosity of AC [38], enhancing the movement of Gr particles [39], which leads to the accumulation of Gr particles in the AC matrix. This effect can be enhanced further when the annealing temperature rises.

Similarly, high-temperature annealing is also helpful to the interconnection of the aggregates, forming more conductive channels, thus enhancing the conductance of the conductive composites [40]. Therefore, the annealing process induces the formation of conductive channels with better stability in the Gr–AC composite, which shrinks the difference of parallelly prepared devices in conductivity caused by the random dispersion of the conductive fillers during preparation. In this case, the higher the annealing temperature, the more uniform the distribution of conductive channels in the Gr–AC composite, while a better uniformity of the device’s performance can be obtained. However, it should be noted that a too-high annealing temperature might lead to the weakening of the on–off control effect of the conductive channel when the conductive channel is subject to the thermal expansion of the AC matrix [41]. Considering the effect that annealing temperature plays on the TCR value and device performance’s uniformity, we chose 100 °C as the optimum temperature to prepare the temperature sensors.

The repeatability and stability are two additional important parameters for temperature sensors, which are supposed to be highly related to the selection of polymer and conductive fillers. The general ways to improve the repeatability and stability of devices are using a multi-component polymer to ensure the integrity of physical materials [26,42] or multi-component conductive fillers to form an interpenetrating network structure [43]. Herein, we compared the repeatability and long-term stability of the devices before and after annealing. Figure 5a shows the resistance versus temperature curves of the devices without annealing after undergoing 1, 50, 100, 150, and 200 heating cycles, respectively, where the curves show considerable differences. The standard error of every data point was high. As a comparison, the resistance versus temperature curves of the devices after 100 °C annealing in Figure 5b show that the curves retain the same value after undergoing different thermal cycles, and the standard error value of the data point is much lower than those of the devices without annealing. The stability was tested by placing the devices on a 35 °C hotplate for 1 h, and the resistance was measured every 5 min. Figure 5c shows that the average resistance of the two groups of temperature sensors presents almost no fluctuation with continuous heating for 1 h, while the standard error of the devices after annealing is obviously smaller than that before annealing. Therefore, after 100 °C annealing, the repeatability and stability of the prepared temperature sensors are improved, which is convenient for the subsequent practical application based on conductive composite materials.

Based on the abovementioned results, we designed a temperature-testing band with an addressable sensor array containing 2 × 9 devices, as shown in Figure 6a, where the magnified diagram of a single device circuit is shown in Figure 6b. The average resistance of the sensors was measured to be 3.1 × 10^3^ Ω at 30 °C after annealing, and the average standard error was 206 Ω, which exhibited good uniformity. As shown in Figure 6c, to explore the potential application of the devices in human-body-temperature detection further, bending tests were executed by winding the band around a plastic cylinder with a diameter of 40 mm. The results are presented in Figure 6d; the band does not show a significant change to the temperature response after bending. Unfortunately, the thickness (78 μm) and size (132 mm × 108 mm) of the band were too large to ensure that all the sensors could be well-attached to the human body for temperature measurements. Moreover, it is worth noting that high-Gr-doping concentration partly reduced the flexibility of the composite, and the nonlinear response of the resistance to temperature increased the difficulty of the application of the corresponding temperature sensors [44]. However, these problems can be solved by optimizing device-preparation technology, replacing Gr particles with metal dendritic conductive fillers [33] and computer-aided modeling of the resistance versus temperature curves [25]. We believe that the flexibility and temperature map testing ability of the band made it capable of accurately measuring the temperature of multiple parts of the human body, such as the back, arm, and leg.

## 4. Conclusions

In conclusion, we studied the performance changes in the temperature sensors based on a Gr–AC conductive composite by varying the annealing temperature, and observed that the TCR values and uniformity of the devices could be improved by thermal annealing, where the best performance was obtained when it was annealed at 100 °C for 30 s. The annealing resulted in the rearrangement of graphite fillers in the polymer. Meanwhile, the cyclic as well as long-term stability properties were improved further after annealing. This result provides a new strategy to solve the problems of low-yield and poor-repeatability flexible temperature sensors in high-volume applications.

## Figures and Tables

**Figure 1 materials-16-01491-f001:**
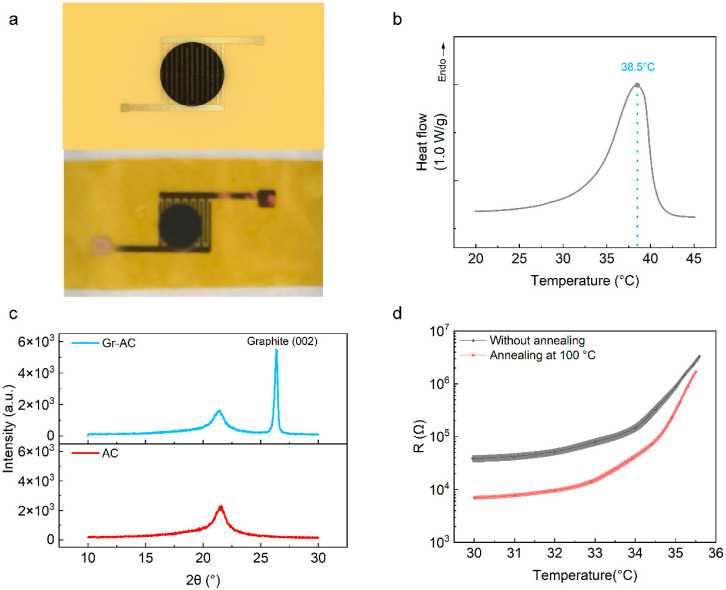
(**a**) Schematic diagram and optical photograph of a single temperature sensor based on a graphite–acrylate copolymer (Gr–AC) composite. (**b**) Differential scanning calorimetry (DSC) measurement results of the Gr–AC composite. (**c**) X-ray diffraction (XRD) results of Gr–AC and AC with the same copolymer mass. (**d**) The resistance versus temperature curves of 18 devices before and after 100 °C annealing for 30 s, where the error bars represent the standard error.

**Figure 2 materials-16-01491-f002:**
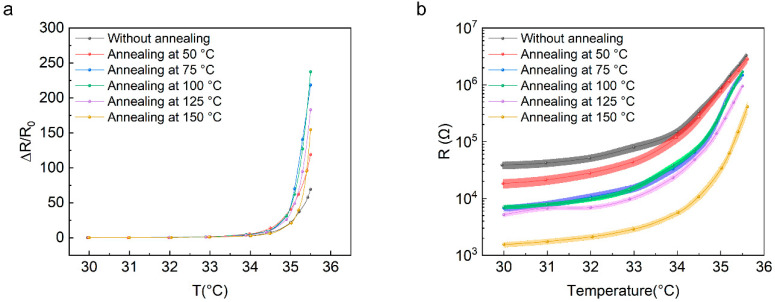
(**a**) The temperature coefficient of resistance (TCR) value and (**b**) resistance versus temperature curves of 18 temperature sensors based on Gr–AC composite before and after annealing at 50, 75, 100, 125, and 150 °C, respectively. The error bars represent the standard error.

**Figure 3 materials-16-01491-f003:**
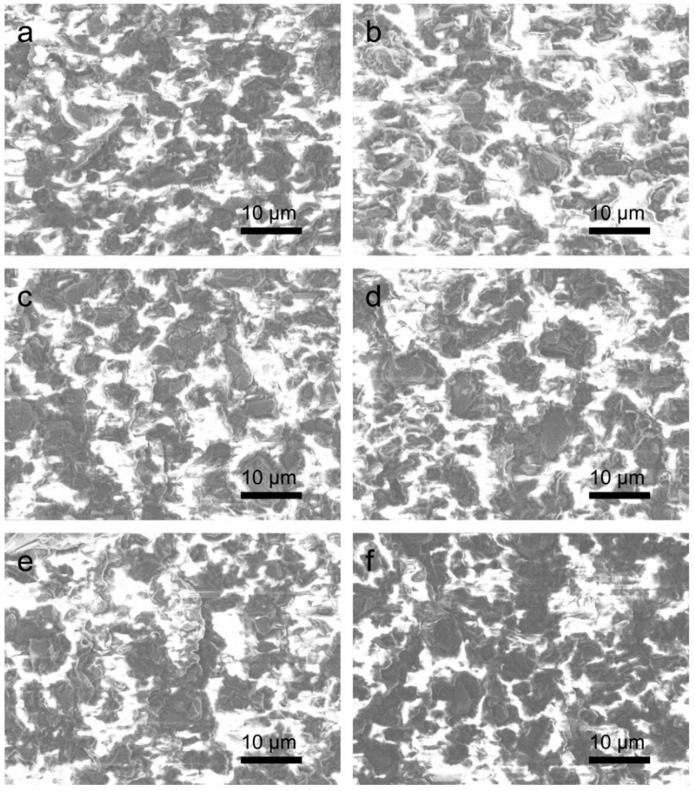
Cross-sectional scanning electron microscopy (SEM) image of Gr–AC conductive composite (**a**) before annealing and annealing at (**b**) 50, (**c**) 75, (**d**) 100, (**e**) 125, and (**f**) 150 °C, respectively.

**Figure 4 materials-16-01491-f004:**
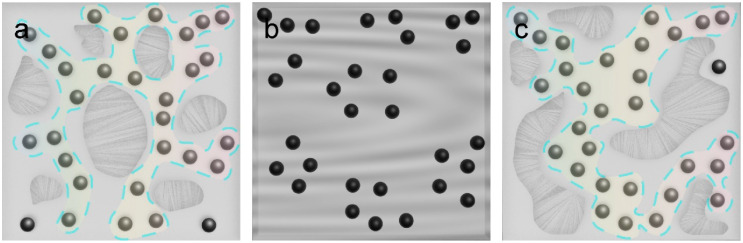
Schematic illustration of the distribution of graphite (Gr) fillers in acrylate copolymer (AC): (**a**) at room temperature, (**b**) at a temperature that is higher than its melting point, and (**c**) re-cooled to room temperature. Possible conductive channels are highlighted in the images.

**Figure 5 materials-16-01491-f005:**
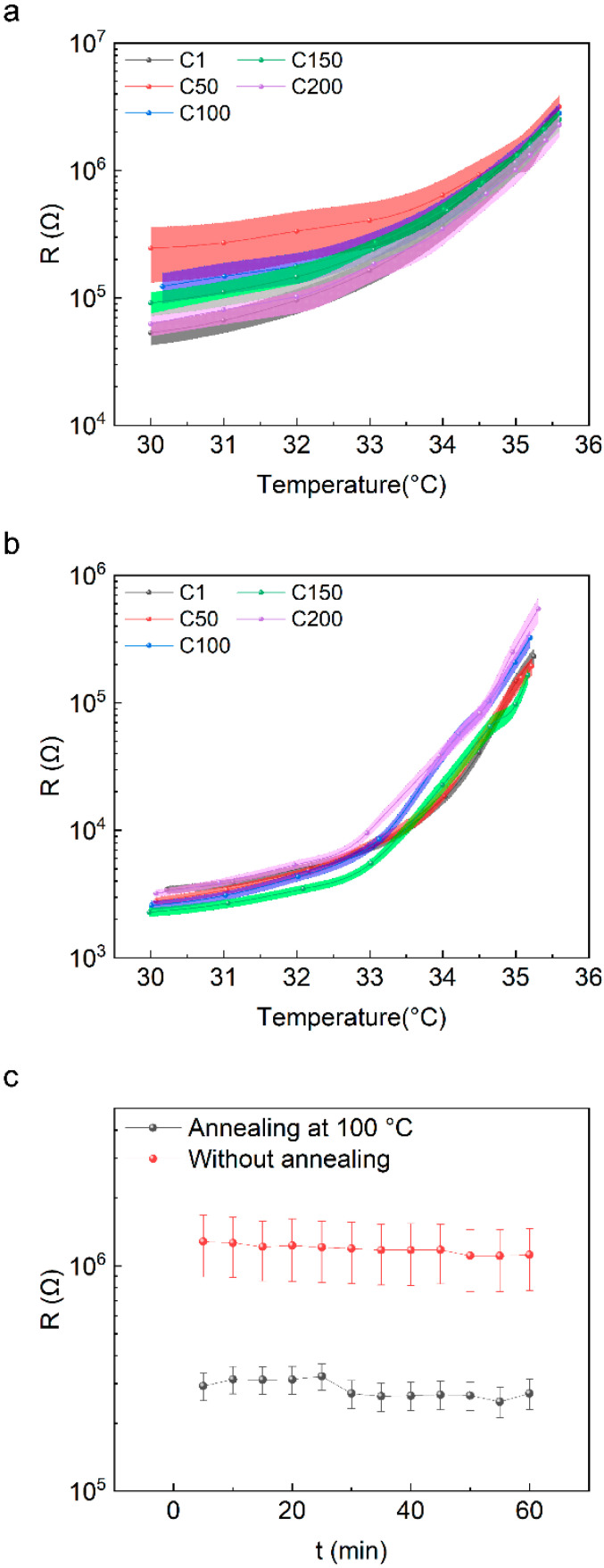
Cyclic stability of 18 temperature sensors (**a**) before and (**b**) after annealing at 100 °C, respectively. (**c**) Continuous measurement of the resistance when 18 temperature sensors were heated at 35 °C for 1 h. The error bar represents the standard error.

**Figure 6 materials-16-01491-f006:**
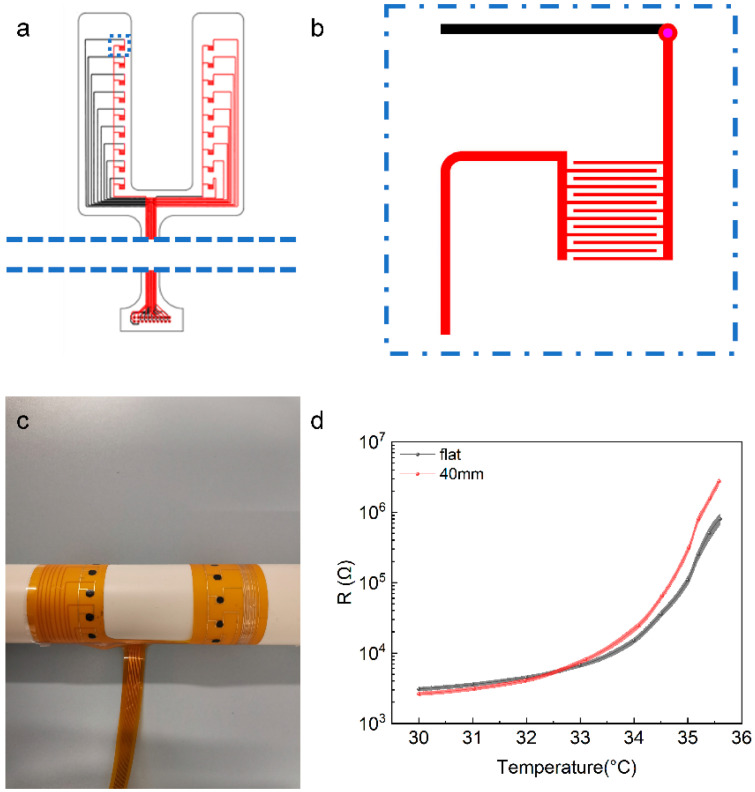
Schematic illustration of **(a**) the addressable circuit and (**b**) a single component of the temperature-testing band. (**c**) Optical photograph of the temperature-sensing array stuck onto a plastic tube with a diameter of 40 mm. (**d**) Bending test of the temperature-testing band.

## Data Availability

Not applicable.

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
