# Peer review of "Fabrication of Temperature Sensors with High-Performance Uniformity through Thermal Annealing"

_materials, 2023, doi:10.3390/ma16041491_

Round 1
Reviewer 1 Report
Review of article:
Authors: Hongrui Liu at all. „ Fabrication of temperature sensors with high performance uniformity through thermal annealing “
Materials MDPI 2022
The temperature of human body is important parameters which is related to diseases of human as well as occurrence and cure diseases. Therefore, the most authors focus their interest on development of flexible temperature sensors allowed temperature measurement not only a tone point as mercury thermometer but on different place on human body.
The authors described in article the principle of flexible thermal sensors which used the different material and compared their parameters. They show, that annealing at temperature 1000C their suggest material with graphite nanoparticles improved parameters TCR, homogeneity, stability, and reproducibility of temperature measurement. The results are presented in graph dependences of resistor on temperature in range 300C – 360C with good sensitivity.
My question to the authors: Why did not measured all present parameters for higher temperature than 360C?
The article takes important problem of human diseases a treatment and possibility to improve it. I recommend this article for publication.
Author Response
Q1: My question to the authors: Why did not measured all present parameters for higher temperature than 36 ℃?
R1: Thank you for this valuable recommendation. Since the upper limit of resistance measurement of the source meter we used is 2 × 108 Ω, the resistance of some devices might be out of range of the instrument if the temperature is above 36 ℃. Accordingly, we revise the manuscript as following:
“In this work, a maximum temperature of 35.5 ℃ is chosen to ensure that all resistances are within the measuring range of the instrument since the upper limit of resistance measurement of the source meter we used is 2 × 108 Ω.” (Page 6 line 14)
Reviewer 2 Report
1- The annealing time is 30 s. Why? Did you try other times? What happens when it increases or decreases?
2- Figure 6 (a,b,c) ranges of y-axis should be modulated to separate the curves more clearly.
3- I suggest that figures (S1, S2 and S5) should be concluded inside the manuscript.
4- Why did not you test your sensor on human?
5- Where are experimental comparisons with other sensors (by data)?
6- More references have to be added to the discussion section.
Reviewer 3 Report
Manuscript ID: materials-2178306
Title: Fabrication of temperature sensors with high performance uniformity through thermal annealing
Authors are required to address these comments for the improvement of their paper.
· What are the stability and repeatability factors of prepared composites? And what were the sample numbers used for this investigation?
· Line number 94 “where both XRD patterns…. ” How do authors come to know the peaks are crystalline AC?
· Figure 4 needs improvement and highlights what it indicates, particularly matrix vs. fillers.
· What are the other limitations that need to be included? Explain how the sample's semicrystalline and crystalline nature affects its sensitivity.
· 4. Experimental sections 4.1. to 4.4., before 2. results and discussion, need to be replaced. Modify heading numbers accordingly.
Round 2
Reviewer 3 Report
the revised manuscript can be accepted.